# Room-Temperature Solution-Processed 0D/1D Bilayer Electrodes for Translucent CsPbBr_3_ Perovskite Photovoltaics

**DOI:** 10.3390/nano11061489

**Published:** 2021-06-04

**Authors:** Bhaskar Parida, Saemon Yoon, Dong-Won Kang

**Affiliations:** School of Energy Systems Engineering, Chung-Ang University, Seoul 06974, Korea; bhaskar.parida@gmail.com (B.P.); saemony@cau.ac.kr (S.Y.)

**Keywords:** translucent, perovskite, solar cell, indium tin oxide, silver nanowires, nanoparticles

## Abstract

Materials and processing of transparent electrodes (TEs) are key factors to creating high-performance translucent perovskite solar cells. To date, sputtered indium tin oxide (ITO) has been a general option for a rear TE of translucent solar cells. However, it requires a rather high cost due to vacuum process and also typically causes plasma damage to the underlying layer. Therefore, we introduced TE based on ITO nanoparticles (ITO-NPs) by solution processing in ambient air without any heat treatment. As it reveals insufficient conductivity, Ag nanowires (Ag-NWs) are additionally coated. The ITO-NPs/Ag-NW (0D/1D) bilayer TE exhibits a better figure of merit than sputtered ITO. After constructing CsPbBr_3_ perovskite solar cells, the device with 0D/1D TE offers similar average visible transmission with the cells with sputtered ITO. More interestingly, the power conversion efficiency of 0D/1D TE device was 5.64%, which outperforms the cell (4.14%) made with sputtered-ITO. These impressive findings could open up a new pathway for the development of low-cost, translucent solar cells with quick processing under ambient air at room temperature.

## 1. Introduction

Translucent solar cells are attracting attention as they have many functional advantages along with clean energy production. They can be applied to transparent windows [1,2] and for building integrated photovoltaic applications [3] for electricity production and have shown great technological growth in recent years. Studies are being conducted on how translucent solar cells can be implemented in a variety of materials, such as dye-sensitized solar cells [4,5], organic solar cells [6,7,8,9], and perovskite solar cells [10,11,12,13,14]. Among them, research using perovskite material is quite active, resulting from the ease of the process, low processing cost, and high power conversion efficiency (PCE) through the solution processing of perovskite precursor materials.

In order to make perovskite solar cells translucent, the rear electrodes, as well as perovskite and charge transport layers, must be transparent. In order to reduce costs, perovskite semiconductors and charge transport layers are generally prepared with solution processing, while front and rear electrodes (i.e., metal oxides) are often made by employing vacuum processes. In addition, heat treatment is included to ensure proper transmittance and electrical conductivity, which causes an unwanted increase in processing steps and costs. Recent interesting research has shown that it requires a pulsed-chemical vapor deposition [15] or sputtering process [16,17,18] to fabricate their oxide-based electrodes. Furthermore, a complex structure of 2D/1D/2D utilizing reduced graphene oxide and Ag nanowire has been developed to provide suitable optoelectronic properties in semitransparent perovskite solar cells [19].

In this work, we develop a rear transparent electrode (TE) that does not require a vacuum process and allows a solution process at room temperature, which can support future industrialization of transparent solar modules with competitive cost. Herein, TEs with 0D/1D duplex structure were developed by stacking indium-tin-oxide nanoparticle (ITO-NPs) and silver nanowires (Ag-NWs) for the first time, based on a room temperature process under an ambient air condition. This electrode was then applied to highly stable CsPbBr_3_ inorganic perovskite solar cells to evaluate their performance. As a result, CsPbBr_3_ cells manufactured through the conventional ITO sputtering process showed a rather low efficiency of 4.14% due to plasma damage, while our suggested 0D/1D bilayer structure contributed to achieving the best PCE of 5.64% without such damage to underlying active layers. Furthermore, the cell with ITO-NPs/Ag-NWs electrode showed a visible light transmittance of 41.6%, approximately similar to that (41.5%) of the device employing the vacuum-processed, standard ITO electrode. These impressive results open up the new possibility of developing high-performance, translucent, low-cost solar modules with whole solution processing at room temperature.

## 2. Materials and Methods

### 2.1. Materials

Indium doped tin oxide (ITO)–coated glass substrates (10 Ω sq^−1^) were purchased from AMGlass, Inc., Miami, FL, USA. Tin oxide (SnO_2_; 15 wt.% in an H_2_O colloidal dispersion), lead bromide (PbBr_2_; ultra–dry, 99.999%), and cesium bromide (CsBr; ultra–dry, 99.9%, metals basis) were purchased from Alfa Aesar, Tewksbury, MA, United States and used without any treatment. Poly (3-hexylthiophene-2,5-diyl) (P3HT) was purchased from Rieke Metals, Lincoln, NE, United States. N,N-dimethylformamide (DMF; 99.5%) was provided from Samchun Chemicals, Soeul, Korea. Chlorobenzene (CBZ; 99.8%) was purchased from Kanto Chemical, Chuo-ku, Japan, and methyl alcohol (99%) was purchased from Acros Organics, Geel, Belgium and used without further purification. The dispersed ITO-NPs (average particle size: 18 nm, 35 wt.% in ethanol) were provided from US Research Nanomaterials, Inc., Houston, TX, United States. Furthermore, Ag-NWs dispersion (diameter: ~30 nm, lengths: ~25 um, 0.5 wt.% in ethanol) were prepared from OK International, Inc., Cypress, CA, USA. Commercial ITO glass substrates for reference were provided from AMGlass, Inc., Miami, FL, USA.

### 2.2. Fabrication of Solution-Processed Transparent Electrode (TE)

The ITO-NPs were spin-coated on the substrates at 3000 rpm for 30 s, followed by coating Ag-NWs under the same conditions. There is no thermal annealing condition for these solutions’ processed translucent electrode layers. All fabrication processes were carried out in ambient air under a relative humidity of 30–40%.

### 2.3. Device Fabrication

ITO glass substrates (2 cm × 2 cm) were cleaned sequentially with acetone and isopropyl alcohol in an ultrasonic system for 20 min. After drying in an oven, the substrates were subjected to an ultraviolet-ozone (O_3_) treatment for 20 min to increase their surface wettability. To deposit SnO_2_ as electron transport layer (ETL), tin oxide dispersion (3 wt.% diluted in DI-water) was spun coated on cleaned ITO substrates at 3000 rpm for 30 s and annealing at 170 °C for 30 min in ambient air. The CsPbBr_3_ perovskite layer was prepared by the two-step solution process. First, The PbBr_2_ solution (1 mmol mL^−1^ in DMF, pre-heated at 70 °C) was spin-coated onto an SnO_2_ layer at 2000 rpm for 30 s and drying at 70 °C for 20 min. Second, the PbBr_2_ substrates were dipped for 25 min in a warmed (40 °C) CsBr methanolic solution (15 mg mL^−1^) and rinsed with methanol. After the rinsing step, the substrates were dried using N_2_ blowing and annealed at 250 °C for 10 min to obtain fully crystallized CsPbBr_3_. After cooling to room temperature (RT), P3HT (10 mg mL^−1^ in CBZ) was spin-coated onto a CsPbBr_3_ film at 3000 rpm for 30 s and annealing at 100 °C for 5 min. For opaque devices, 150 nm Au electrodes were deposited under a pressure of 5 × 10^6^ Torr using a thermal evaporator by defining an effective area of 4 mm^2^ with a metal aperture. Regarding translucent cells made with vacuum process for comparison, sputtered ITO films were prepared at room temperature under the identical base pressure. As for the solution-processed translucent cell, ITO-NPs/Ag-NWs were made with the above condition as the film preparation.

### 2.4. Measurements and Characterization

The optical absorbance of the films was characterized using ultraviolet-visible (UV-vis.) spectroscopy (UV-2700, Shimadzu, Japan). The sheet resistance of the translucent electrodes was measured using a 4-point probe system (M4P 302 Function, MS Tech., Hwaseong, South Korea) connected with Source Measure Unit (SMU; Keithley 2400, Cleveland, OH, United States). The cross-section image of the translucent perovskite solar cells was observed by employing field-emission scanning electron microscopy (FE-SEM; AURIGA, Carl Zeiss, Jena, Germany). For device characterization, the current density-voltage (J–V) characteristics were measured using a solar simulator (PEC–L01, Peccell Technologies, Yokohama, Japan) under the standard condition of air mass (AM) 1.5G (100 mW cm^−2^) at room temperature.

## 3. Results and Discussion

In order to realize translucent photovoltaic devices, the rear electrode as well as the front electrode should be transparent by replacing the opaque metal electrode. Hence, we believe that controlling the optoelectronic properties of the transparent rear electrode is one of the critical issues to offer translucent characteristics to the solar cells. Therefore, we first prepared the solution-processed electrodes on the glass substrates to study their optical properties. Furthermore, bare glass and sputtered ITO films were also added for references, as revealed in Figure 1a. The room-temperature, solution-processed TEs exhibit quite high transparency comparable to the vacuum processed one. To investigate in detail, Figure 1b compares optical transmissions of various TE films fabricated by solution-processed nanoparticles/nanowires and magnetron sputtering. Furthermore, Table 1 includes transmittance at 550 nm (T_550nm_), average visible transmission in 400–750 nm (AVT), sheet resistance, and figure of merit (FOM) to assess their properties. Room-temperature solution-processed ITO-NPs TE exhibited AVT and sheet resistance (R_s_) of 92.1% and 162.7 kΩ/sq, while those of the sputtered ITO film were 80.2% and 38.5 Ω/sq, respectively. For comparison between solution-processed nanoparticle without annealing and sputtered film in vacuum, such high AVT and high R_s_ of ITO-NPs might be attributed to the rather porous film structure, which is examined in Figure 2. On the contrary, sputtered ITO exhibited much better Rs due to its compact structure, as known. Haacke Figure of merits [20] is helpful to compare the relative performance of transparent conductors. Using FOM, ITO-NPs and sputtered ITO TEs show 3.2 × 10^6^ Ω^−1^ and 2.7 × 10^−3^ Ω^−1^, which implies that only ITO-NPs film made without any annealing provides inferior transparent conducting properties compared to the control ITO by sputtering. In other words, poor conductivity limits the FOM. Therefore, another option to consider is introducing Ag-NWs that exhibited AVT and R_s_ of 96.0% and 9.1 Ω/sq, respectively. As a result, the FOM of the Ag-NWs TE was estimated as 7.7 × 10^−2^ Ω^−1^, which outperforms reference sputtered ITO TE (processed at room temperature) with a magnitude of more than 1 order. In the case of the ITO-NPs/Ag-NWs bilayer TE structure, it exhibits AVT and R_s_ of 84.3% and 27.7 Ω/sq, respectively, providing an FOM of 8.6 × 10^−3^ Ω^−1^. These optoelectronic properties shown in Table 1 were also observed in Figure 1c–e. From UV-Vis spectroscopy, the light absorbance of the various TEs was investigated. Ag-NWs provide the lowest absorbance among the other ones, which can be attributed to random network with many vacant spaces between wire and wire. Here, it is noted that the device quality of ITO-NPs is 1.5 μm in thickness, which can cause free carrier absorption that is shown in long-wavelength regions. In the case of identical thickness of 150 nm, we could observe almost negligible transmission loss in the longer wavelengths as exhibited in Appendix A. Figure 1d displays I–V characteristics of various TEs. From estimating their conductance, ITO-NPs, Ag-NWs, ITO-NPs/Ag-NWs bilayer, and sputtered ITO film showed 0.167 mS, 72.3 S, 47.3 S, and 46.7 S, respectively. Because of the small slope of ITO-NPs, its magnified plot is given in Figure 1e. By adopting Ag-NWs onto ITO-NPs, its conductance is dramatically improved, as is also found in FOM of the Table 1. In Table 1, the specification of the commercial ITO is also added. We have found out the commercial ITO with the best TE performance, from AMG glass Inc. For more information, commercial ITO is made by conventional magnetron sputtering and subsequent post-thermal treatment at 400 °C for sufficient dopant activation and thus conductivity. Therefore, the commercial ITO exhibits better transmission and lower resistance compared with other solution-processed NPs/NWs and/or room-temperature-processed in-house ITO. It should be noted here that our sputtered (in-house) ITO sample was made at room temperature to avoid any thermal damage to the underlying perovskite and organic charge transport layers during the deposition process of rear ITO TE.

Figure 2 exhibits the surface topography of various solution-processed NPs/NWs TEs in order to crosscheck the above properties, analyzed by FE-SEM. As for ITO-NPs, we could find good surface coverage of NPs on the glass substrate as shown in Figure 2a. Furthermore, there are some pinholes and/or vacancies on the surface, which can imply that the film would be rather porous. Figure 2b indicates the well-prepared Ag-NWs, which consist of high–density bundles of Ag-NWs networks that are randomly distributed on the substrate. Furthermore, as revealed in Figure 2c, one can find that the Ag-NWs are successfully formed on the ITO-NPs film, indicating successful fabrication of 0D/1D bilayer architecture to boost electrical properties of room-temperature processed ITO-NPs film. As a comparison, the surface topography of sputtered ITO film was also given in Figure 2d, which shows a very smooth and compact structure because of the well-controlled plasma deposition in a vacuum. Appendix A also display such various transparent films with different scale (2 μm in scale bar) to provide better observation of film structure.

To fabricate 0D/1D bilayer TE, we applied spin coating for the ITO-NPs and Ag-NWs to demonstrate all–solution processing. To confirm its successful stacked architecture, we identified the presence of the different TE films in the bilayer architecture as measured by X-ray diffraction (XRD). In Appendix A, XRD patterns of fabricated three types of solution processed TEs are given. All the diffraction peaks regarding ITO, such as (211), (222), (400), (411), (332), (431), (440), and (622), are marked with # [21,22] for the ITO-NPs and ITO-NPs/Ag-NWs films. In Appendix A, furthermore, the diffraction peak located at 38.2° corresponds to Ag-NWs [23], which were also reflected in ITO-NPs/Ag-NWs. From these XRD observations, it could be confirmed that the proposed ITO-NPs/Ag-NWs stack is well formed based on the continuous spin coating method.

In order to assess those room-temperature-processed NPs/NW TE films, the device structures (ITO/SnO_2_/CsPbBr_3_/P3HT/transparent electrodes) for planar-type inorganic perovskite solar cells employing various rear electrodes were constructed, as depicted in Figure 3a. In addition to the optoelectronic and surface morphological characteristics shown in Figure 1 and Figure 2, the energy band alignment of such transparent electrodes is also crucial for the device operation. As illustrated in Figure 3b, ITO-NPs and Ag-NWs provide favorable band alignment for hole collection from P3HT HTL when illuminated [19]. In the case of the work function of ITO-NPs, which was investigated by UPS measurement, it is shown in Appendix A. After the device fabrication, the cross-sectional image of the device were analyzed by FE-SEM, as displayed in Figure 3c. ETL(SnO_2_) and HTL(P3HT) of each ~20 nm in thickness were too thin to be clearly visible in the image (scale bar = 300 nm). ITO-NPs of about 1.5 μm and Ag-NWs of 160 nm were obviously confirmed on the HTL/perovskite (CsPbBr_3_, ~240 nm). Figure 3d reveals J–V characteristics of fabricated perovskite solar cells with ITO-NPs and ITO-NPs/Ag-NWs bilayer rear TEs. As also summarized in Table 2, the cell with only ITO-NPs exhibit inferior performance, which is attributed to poor fill factor (FF) and short-circuit current (J_SC_). As indicated in the J–V curve with high series resistance, the inferior electrical conductivity of room-temperature-processed ITO-NPs would limit charge carrier collection in rear TE. In addition, as shown in Appendix A, insufficient performance, especially reduction in FF, was also observed when only Ag-NWs were used, which can be ascribed to poor lateral conductivity of P3HT that limits lateral charge transport to be collected at the randomly distributed Ag-NWs. With introducing ITO-NPs/Ag-NWs bilayer TE, however, the fill factor and J_SC_ values are dramatically improved, and thus remarkable enhancement in power conversion efficiency (PCE) from 1.27 to 5.64% was confirmed. This is possibly due to a significant reduction in the resistance as confirmed in Figure 1e and Table 1. Ag-NWs with superior conductivity contributed to providing appropriate lateral conduction of charge carriers, which generates quick carrier collection in the device. As depicted in Figure 3e, additional introduction of Ag-NWs revealed less sacrifice of AVT for the ITO-NPs–coated perovskite cell (46.2→41.6%) in consideration of the superior gain in PCE, thanks to the remarkable transmission property of the Ag-NWs. The inset figure indicates the overall fabricated translucent perovskite solar cells with ITO-NPs/Ag-NWs bilayer that provides good transparency. Based on these results, it also seems interesting to compare this solution-processed device with the device characteristics using rear TE by vacuum deposition.

Figure 4 compares the ITO-NPs/Ag-NWs–coated CsPbBr_3_ solar cell performance with other vacuum-processed devices employing opaque Au and sputtered ITO. Furthermore, detail photovoltaic parameters are summarized in Table 3. The best-performing device using thermally evaporated Au exhibits a high open-circuit voltage (V_OC_) and PCE of 1.32 V and 7.05%, respectively. The wide bandgap (~2.3 eV) of CsPbBr_3_ photovoltaic absorber can be used as a top cell of the perovskite/organic or perovskite/perovskite tandem cells to provide UV filtering (for protecting organic cell) and/or UV-Vis spectra absorption. Because of the Au electrode, this cell reveals opaque properties, as shown in transmission spectra (Figure 5b) and AVT of 0% in Table 3. In the case of the perovskite cell with sputtered ITO TE, it showed lower performance such as V_OC_ (1.15 V) and FF (0.570) compared to those of the cell with Au. This would be ascribed to the typical plasma damage of magnetron sputtering process onto organic P3HT/perovskite structure, as also presented in the literature [18]. Hence, an additional buffer layer using atomic layer deposition was also suggested to avoid such plasma damage [24,25]. As for the proposed cell with ITO-NPs/Ag-NWs TE, however, it shows comparable V_OC_ (1.32 V) with that (1.33 V) of the cell using Au. Even though the PCE (5.64%) of the translucent cell employing ITO-NPs/Ag-NWs is rather low compared with the electrical performance of Au–coated device due to the J_SC_ and FF, this room-temperature-processed cell offers an impressive AVT of 41.6%, which supports totally different, translucent application from a just opaque device such as building-integrated photovoltaics, transparent smart windows, etc. If we compare the device (ITO-NPs/Ag-NWs) with one using sputtered ITO, similar EQE, J_SC_, and AVT were confirmed. On the other hand, the overall performance of the device using ITO-NPs/Ag-NWs is much superior because the PCE of the proposed cell outperforms that of the cell made with sputtered ITO made in vacuum, which is an additional impressive result for translucent photovoltaic application. The suggested device requires no costly vacuum deposition and does not accompany plasma damage during the fabrication process. This electrode also supports a very quick, ambient process without any further post thermal treatment. Statistical evaluation of the fabricated devices is given in Figure 5. The fabricated devices provide decent reproducibility compared to the solar cells with rear electrodes processed in vacuum (Au, sputtered ITO). The detailed PV parameters with statistical data were included in Appendix A. Further optimization to reduce the FF gap between the Au–coated device and ITO-NPs/Ag-NWs–based cell could promote the competitiveness of this translucent solar cells by enhancing the PCE close to the opaque device in addition to providing great optical transmission capabilities.

## 4. Conclusions

In this work, we constructed a bi-layered top TE by employing solution-processable zero- and one-dimensional materials at room temperature without any post-treatment. The 0D ITO-NPs and 1D Ag-NWs stacked structure presented comparable or higher transparent conducting properties compared to sputtered ITO. This 0D/1D supported sufficient carrier collection of devices when used in translucent CsPbBr_3_ perovskite solar cells. The device with the suggested TE architecture reveals an efficiency up to 5.64% with providing average visible transmittance over 41%, which was better than that (4.14%) of costly vacuum-processed ITO TE. We believe that this new 0D/1D TE configuration can be promising for further progress of low-cost, solution-processable translucent perovskite solar cells that are applicable for building integrated photovoltaics, tandem, and portable electronic devices.

## Figures and Tables

**Figure 1 nanomaterials-11-01489-f001:**
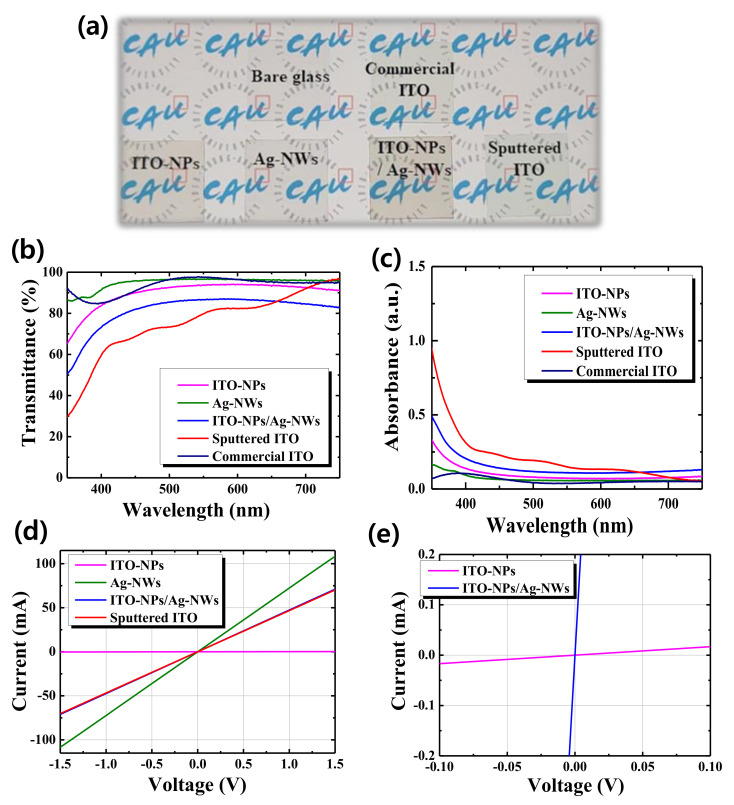
(**a**) An image of samples of various transparent electrodes (TEs) such as ITO nanoparticles (ITO-NPs), Ag nanowires (Ag-NWs), ITO-NPs/Ag-NWs bilayer, and sputtered in–house and commercial ITO films, coated on glass substrates, (**b**) optical transmittance and (**c**) absorbance of the TEs, (**d**) current (I)–voltage (V) characteristics, and (**e**) magnified I–V plot for ITO-NPs and ITO-NPs/Ag-NWs TEs.

**Figure 2 nanomaterials-11-01489-f002:**
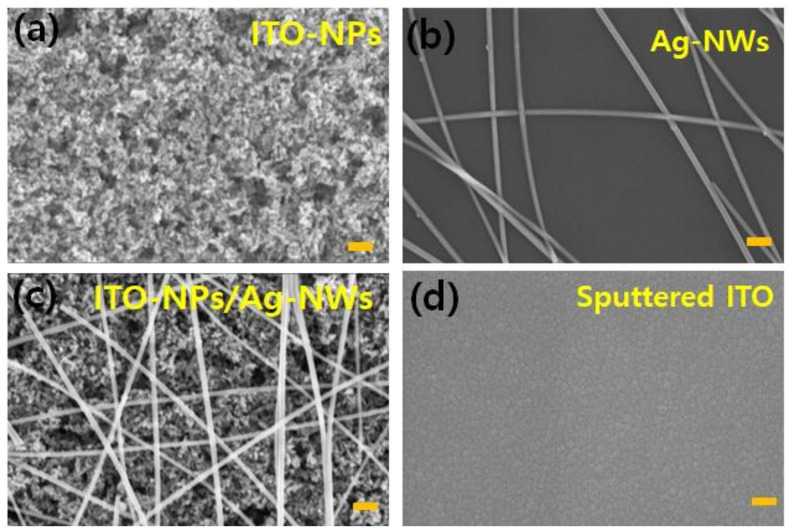
Field enhanced scanning electron microscope (FE-SEM) (**a**) ITO-NPs, (**b**) Ag-NWs, (**c**) ITO-NPs/Ag-NWs bi-layer, and (**d**) sputtered ITO films (scale bar = 200 nm).

**Figure 3 nanomaterials-11-01489-f003:**
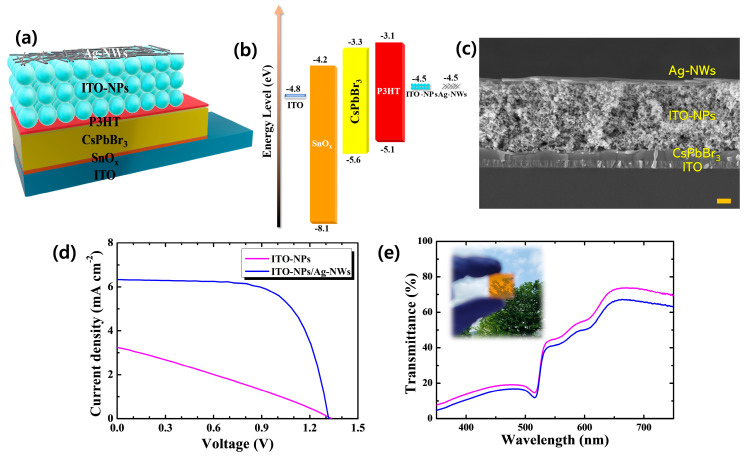
(**a**) Constructed n–i–p planar CsPbBr_3_ perovskite solar cell architecture, (**b**) designed energy band diagram, (**c**) SEM image of the cross–sectional device (scale bar = 300 nm), (**d**) J–V characteristics, and (**e**) device transmittance of fabricated perovskite solar cells with of ITO-NPs, ITO-NPs/Ag-NWs bilayer.

**Figure 4 nanomaterials-11-01489-f004:**
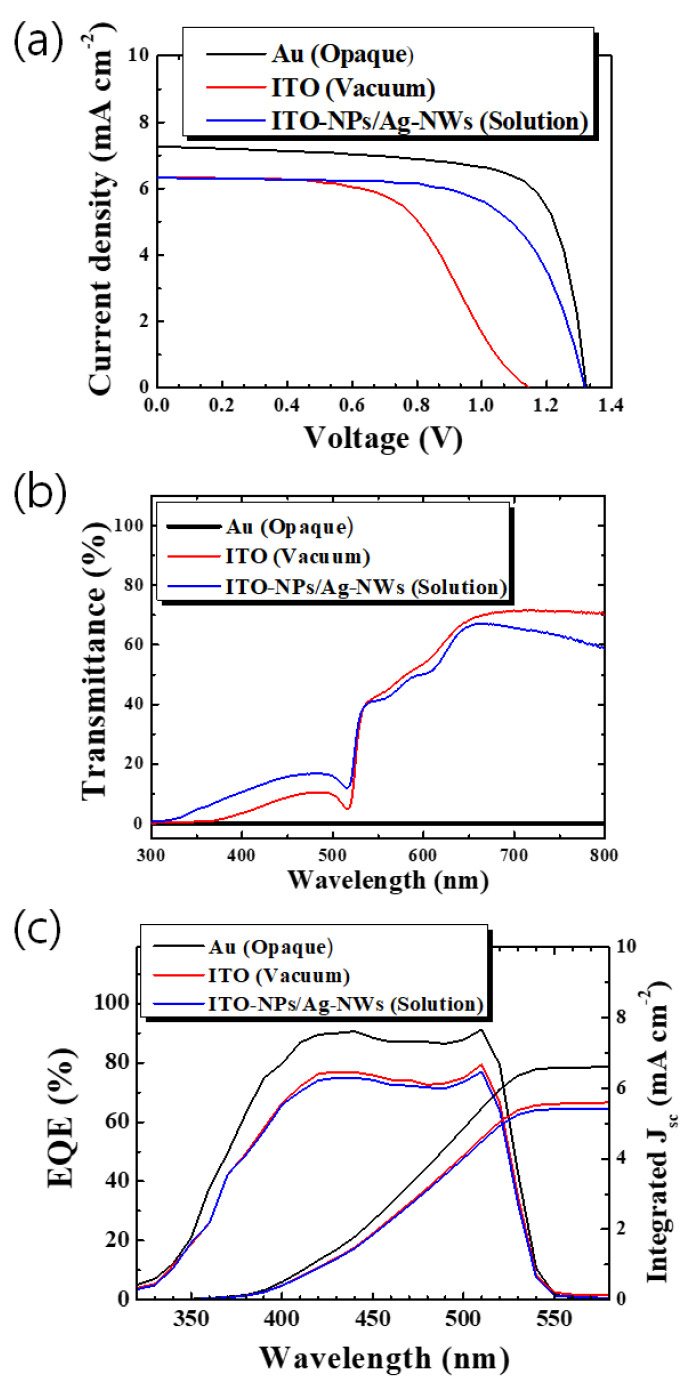
(**a**) J–V characteristics, (**b**) device transmittance, and (**c**) EQE, integrated J_SC_ of fabricated CsPbBr_3_ perovskite solar cells with ITO-NPs/Ag-NWs, vacuum processed ITO, and Au.

**Figure 5 nanomaterials-11-01489-f005:**
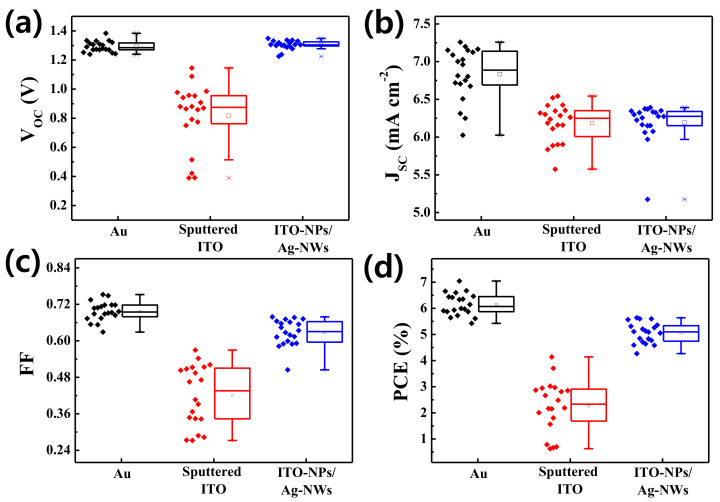
Box charts exhibiting the statistical evaluation (20 samples for each) of (**a**) V_OC_, (**b**) J_SC_, (**c**) FF, and (**d**) PCE of the CsPbBr_3_ perovskite solar cells with ITO-NPs/Ag-NWs, vacuum-processed ITO, and Au.

**Table 1 nanomaterials-11-01489-t001:** Optoelectronic properties of various TEs such as transmittance at 550 nm (T_550nm_), average visible transmission in 400–750 nm (AVT), sheet resistance, and figure of merit (FOM).

Electrodes	ITO-NPs	Ag-NWs	ITO-NPs/Ag-NWs	Sputtered ITO	Commercial ITO
*T*_550 nm_ (%)	93.7	96.6	86.6	79.8	97.6
AVT (%)	92.1	96.0	84.3	80.2	94.5
Sheet resistance (Ω/sq)	162.7 k	9.1	27.7	38.5	10.0
FOM (Ω^−1^)	3.2 × 10^−6^	7.7 × 10^−2^	8.6 × 10^−3^	2.7 × 10^−3^	7.8 × 10^−2^

**Table 2 nanomaterials-11-01489-t002:** Device performance of fabricated CsPbBr_3_ perovskite solar cells with ITO-NPs and ITO-NPs/Ag-NWs rear electrode.

Electrodes	V_OC_ (V)	J_SC_ (mA cm^−2^)	FF	PCE (%)	AVT (%)
ITO-NPs	1.33	3.20	0.299	1.27	46.2
ITO-NPs/Ag-NWs	1.32	6.33	0.677	5.64	41.6

**Table 3 nanomaterials-11-01489-t003:** Device performances of translucent cell with ITO-NPs/Ag-NWs and vacuum-processed ITO. Conventional opaque device using Au electrode is given for better comparison.

Electrodes	V_OC_ (V)	J_SC_ (mA cm^−2^)	FF	PCE (%)	AVT (%)	EQE J_SC_ (mA cm^−2^)
Au (Opaque)	1.32	7.26	0.735	7.05	−	6.62
ITO (Vacuum)	1.15	6.35	0.570	4.14	41.5	5.67
ITO-NPs/Ag-NWs (Solution)	1.32	6.33	0.677	5.64	41.6	5.46

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
