# Peer review of "Room-Temperature Solution-Processed 0D/1D Bilayer Electrodes for Translucent CsPbBr3 Perovskite Photovoltaics"

_nanomaterials, 2021, doi:10.3390/nano11061489_

Round 1

Reviewer 1 Report

This manuscript introduced  TE based on ITO nanoparticles by solution processing in ambient conditions. As it reveals insufficient conductivity, Ag nanowires (Ag-NWs) are additionally coated. The ITO-NPs/Ag-NW (0D/1D) bilayer TE exhibits better figure of merit than sputtered ITO. After constructing CsPbBr3 perovskite solar cells, the device with 0D/1D TE offers similar average visible transmission with the cells with sputtered ITO. More interestingly, power conversion efficiency of 0D/1D TE device was 5.64% which outperforms the cell (4.14%) made with sputtered-ITO. The work is innovative , but there are still some problems.It is therefore recommended to be revised for publication.

1. In Figure b and c, compared with ITO nanoparticles (ITO-NPs), Ag nanowires (Ag-NWs) and sputtered ITO, the transmittance of ITO-NPs/Ag-NWs bilayer is obviously reduced in red wavelengths and the absorption is obviously enhanced in red wavelengths. Why is this?

2. Ag nanowires has better transmittance and conductivity. What is the change of device performance compared with Ag nanowires and ITO-NPs/Ag-NWs?

3. Device performances of translucent cell with ITO-NPs/Ag-NWs and vacuum processed ITO: compared with ITO(Vacuum), why does the VOC of ITO-NPs/Ag-NWs(Solution) increase so much?

4. Compared with ITO(Vacuum), the transmittance of ITO-NPs/Ag-NWs increases. why does the current density decrease instead of increase?

5. The ITO-NPs layer is significantly thicker than the other layers. What effect does this have on the battery performance and light transmittance?

6. The performance of ITO-NPs/Ag-NWs -based translucent cells is significantly worse than that of ordinary perovskite solar cells, and is it possible to improve the performance?

Author Response

Please see the attachment " Revision note"

Thank you so much.

Reviewer 2 Report

The work provided by Parida et al. introduces a transparent electrode based on ITO nanoparticles and Ag nanowires that, despite being not novel in the field of optoelectronics, it provides interesting results for device processing in ambient air. I would recommend its publication after some revisions are provided:

  • The FOM, transmittance and resistivity of the different electrodes are provided in Figure 1 and Table 1. However, the comparison is only provided for the ITO materials prepared by the authors, using as reference a sputtered ITO film. A more realistic comparison should include the data from a commercial ITO reference, as the one the authors use as the back transparent electrode in their devices. Therefore, they should include a reference commercial ITO in all panels of Figure 1 and Table 1.
  • In the Fabrication section, the authors describe the preparation of the transparent electrode by a sequential desposition of ITO-NPs and Ag-NWs. Given that there is no thermal treatment, I wonder how it is possible that using the same solvent, part of the first material is not removed during the deposition of the Ag-NWs. Similarly, there is no data regarding the materials characterization of the different layers. It would be interesting to add the XRD analysis of the transparent electrode in SI, to identify the presence of the different components.
  • The authors should make homogeneous their graphs. They use  different colours and line-shapes in each panel for the materials, which makes it more difficult to compare the results. They should correct it.
  • The fabrication and composition of one specific material strongly affects its energetic profile. In Figure 3b, the authors provide the energy level diagram used in this manuscript. However, they assume the value for their prepared TE’s is the same between them as well as to commercial ITO. I recommend them to verify the band positions (work function) of their several TE by proper experimental techniques.
  • The statistical analysis should also indicate the number of samples per condition (and in total) employed for the analysis.

Author Response

(The authors gave the same response as above.)
